

# Writer verification of partially damaged handwritten Arabic documents based on individual character shapes

Majid A. Khan, Nazeeruddin Mohammad, Ghassen Ben Brahim, Abul Bashar and Ghazanfar Latif

College of Computer Engineering and Science, Prince Mohammad Bin Fahd University, Khobar, Eastern Province, Saudi Arabia

## ABSTRACT

Author verification of handwritten text is required in several application domains and has drawn a lot of attention within the research community due to its importance. Though, several approaches have been proposed for the text-independent writer verification of handwritten text, none of these have addressed the problem domain where author verification is sought based on partially-damaged handwritten documents (*e.g.*, during forensic analysis). In this paper, we propose an approach for offline text-independent writer verification of handwritten Arabic text based on individual character shapes (within the Arabic alphabet). The proposed approach enables writer verification for partially damaged documents where certain handwritten characters can still be extracted from the damaged document. We also provide a mechanism to identify which Arabic characters are more effective during the writer verification process. We have collected a new dataset, Arabic Handwritten Alphabet, Words and Paragraphs Per User (AHAWP), for this purpose in a classroom setting with 82 different users. The dataset consists of 53,199 user-written isolated Arabic characters, 8,144 Arabic words, 10,780 characters extracted from these words. Convolutional neural network (CNN) based models are developed for verification of writers based on individual characters with an accuracy of 94% for isolated character shapes and 90% for extracted character shapes. Our proposed approach provided up to 95% writer verification accuracy for partially damaged documents.

Corresponding author
Majid A. Khan, makhan@pmu.edu.sa

## INTRODUCTION

Handwriting is a skill that most people develop over the years and is considered a behavioral distinguishing factor among individuals (*Rehman, Naz & Razzak, 2019*). It is unlikely that two different individuals will produce very similar handwriting (*Srihari et al., 2002*) and therefore handwriting can be used for forensic analysis by domain experts and could be of major importance in the process of identifying authorship of documents, signatures forgery, alteration detection, legal documents verification, *etc*. The differences in people's handwriting are most likely to manifest and be very noticeable when the considered writing

language is with many variations in terms of the language dimension such that the number of existing characters, their shapes and deviations when appearing in words compared to appearing in sentences or even when being isolated characters.

The Arabic language has lately been the focus of much research due to its widespread use as well as the inherent challenges in terms of being a complex character-based language. All Arabic language related research can be categorized into any of these areas, namely: character recognition, writer identification and verification, text-to-speech conversion, speech recognition, language analysis, understanding and translation (*Rehman, Naz & Razzak, 2019*). Most of the research work has attempted to deal with the challenges encountered with the nature of the Arabic language. These challenges can be summarized in the following four items:

- Alphabet characters large variations – the number of characters along with their variations in terms of their positions in words (isolated, initial, end, and middle) include 101 different shapes (*Ahmed & Sulong, 2014*). Figure 1 shows all possible variations of all 28 characters in the Arabic alphabet. It is worth mentioning that the same table could be augmented with three composed special Arabic character (Arabic long vowels "alif (ا)", "waw (و)" and "yaa (ي)" with a "hamza (ء)" being placed on top or bottom of the character. This makes the total number of character variations reach 111.
- Character similarities – many of the characters are very similar in shape, however, the only difference may be the position of a single "dot" or the number of dots.
- Human writing style – differs from one individual to another in terms of character shapes, size, overlap, and how neighboring characters are being interconnected. For instance, one individual may write multiple dots as a connected line segment, while others may write them separately.
- Arabic language cursive nature – in the sense that there exists a "virtual" baseline that connects words when writing sentences. This cursive nature distinguishes the Arabic language from others (such as Latin, Chinese, *etc.*).

All of the aforementioned challenges have made the problem of recognizing individuals based on their handwriting very appealing, perplexing, and important to the research community in their support to forensic science. Handwriting-based individual recognition could be classified into two sub-categories: (1) verification and (2) identification. The former is considered as a binary classification problem which involves the decision of rejecting or accepting the authentication of a handwriting sample with other samples. On the other hand, the latter is a multinomial classification which attempts to identify a genuine writer among a list of many writers based on handwriting similarities.

Extensive research has been done related to the topic of identifying authors based on existing handwriting and numerous approaches have been suggested to handle such a problem. Most of the state of the art approaches were based on the analysis of words or sub-words from handwritten Arabic scripts (*Maliki, Al-Jawad & Jassim, 2017*). They have attempted to create feature vector following a manual feature extraction process; a step deemed difficult since it requires Arabic knowledge and expertise to ensure that the influential features are being targeted and eventually extracted (*Rehman et al., 2019*).

| No. | Group | Character | Character Shape Variations | | | | No. of different forms |
|-----|-------|-----------|---------|-------|--------|-----|------------------------|
|     |       |           | Regular | Begin | Middle | End |                        |
| 1 | 1 | Alif | ا | ا | ـا | ـا | 2 |
| 2 |   | Beh | ب | بـ | ـبـ | ـب | 4 |
| 3 | 2 | Teh | ت, ة | تـ | ـتـ | ـت, ـة | 6 |
| 4 |   | Theh | ث | ثـ | ـثـ | ـث | 4 |
| 5 |   | Jeem | ج | جـ | ـجـ | ـج | 4 |
| 6 | 3 | Haa | ح | حـ | ـحـ | ـح | 4 |
| 7 |   | Khah | خ | خـ | ـخـ | ـخ | 4 |
| 8 | 4 | Dal | د | د | ـد | ـد | 2 |
| 9 |   | Thal | ذ | ذ | ـذ | ـذ | 2 |
| 10 | 5 | Raa | ر | ر | ـر | ـر | 2 |
| 11 |   | Zay | ز | ز | ـز | ـز | 2 |
| 12 | 6 | Seen | س | سـ | ـسـ | ـس | 4 |
| 13 |   | Sheen | ش | شـ | ـشـ | ـش | 4 |
| 14 | 7 | Sad | ص | صـ | ـصـ | ـص | 4 |
| 15 |   | Dad | ض | ضـ | ـضـ | ـض | 4 |
| 16 | 8 | Tah | ط | طـ | ـطـ | ـط | 2 |
| 17 |   | Thah | ظ | ظـ | ـظـ | ـظ | 2 |
| 18 | 9 | Ain | ع | عـ | ـعـ | ـع | 4 |
| 19 |   | Ghain | غ | غـ | ـغـ | ـغ | 4 |
| 20 | 10 | Feh | ف | فـ | ـفـ | ـف | 4 |
| 21 | 11 | Qaf | ق | قـ | ـقـ | ـق | 4 |
| 22 | 12 | Kaf | ك, ك | كـ | ـكـ | ـك, ـك | 6 |
| 23 | 13 | Lam | ل | لـ | ـلـ | ـل | 4 |
| 24 | 14 | Meem | م | مـ | ـمـ | ـم | 4 |
| 25 | 15 | Noon | ن | نـ | ـنـ | ـن | 4 |
| 26 | 16 | Heh | هـ, ه | هـ | ـهـ | ـه | 5 |
| 27 | 17 | Waw | و | و | ـو | ـو | 2 |
| 28 | 18 | Yaa | ي | يـ | ـيـ | ـي | 4 |

**Figure 1** Character shapes in Arabic alphabet grouped per similarity in writing style.

It is shown that the performance of the writer identification model is highly dependent on the selection of features along with the applied classifier (*Rehman et al., 2019*). Writer identification using handwriting approaches can be categorized into two broad categories: (1) text-dependent and (2) text-independent (*Xing & Qiao, 2016*). Earlier text-dependent approaches using words for writer identification (or verification) have focused on learning from a small set of user written words. Although this approach works quite well on these

**Figure 2** **Writer verification for partially damaged document.**

selected words, it is difficult to scale to include all possible words and their variants in the Arabic dictionary.

Existing research work in writer identification and verification domain has primarily focused on identifying techniques to determine authorship assuming that an undamaged user writing is available for the task at hand (*Abdi & Khemakhem, 2015*; *Ahmed & Sulong, 2014*; *Rehman, Naz & Razzak, 2019*). These techniques have not considered the likely scenario in forensic analysis where user identification or verification is sought based on a partially damaged documents as shown in Fig. 2. This has increased the complexity of such task as it is difficult for machine learning based models to generalize for all possible distortions in handwritten documents.

In this paper, we propose a writer verification approach based on individual Arabic character shapes. The proposed approach enables writer verification for partially damaged documents where certain handwritten characters can still be extracted from the damaged document. This approach also has the additional advantage that the set of Arabic character shapes is limited and a deep learning model can be easily trained on a complete set of characters as opposed to considering an unreasonably large Arabic word-based dataset. The writer verification can thus be performed by extracting character shapes from the undamaged part of the document and then using the learned model to identify/verify the user. It is important to mention that the proposed approach is not dependent on specific Arabic words, works very well for any word in the Arabic dictionary, and is not limited by the number of unique words captured in the dataset.

Our proposed approach requires a dataset of user handwritten Arabic characters per user. Unfortunately, existing Arabic writer identification datasets did not either provide user handwritten characters (rather contain only words or sentences) such as in famous

IFN/ENIT, KHATT, QUWI datasets, or did not provide user information such as in Hijja, AHCD datasets. Therefore, we have collected a new dataset named Arabic Handwritten Alphabet, Words and Paragraphs Per User (AHAWP), which contains handwritten characters, words and sentences from 82 different users in a classroom setting (*Khan, 2022*). The dataset is publicly available at: https://data.mendeley.com/datasets/2h76672znt/1.

Our proposed approach has attempted to address the following research questions:

- How effective are the CNN-based architectures to learn a model that can verify users based on their handwritten Arabic character shapes?
- How accurately can we use the models trained on these individual character shapes to verify users based on Arabic character shapes extracted from the partially damaged handwritten document?

## Paper contributions

The key contributions of this paper can be summarized as follows:

- Proposes and evaluates an individual character based text-independent writer verification approach for partially-damaged handwritten Arabic documents.
- Provides a comprehensive Arabic language dataset, which consists of 53,199 handwritten isolated Arabic characters, 8,144 Arabic words (which encompass all characters), and 10,780 character shapes extracted from these words. The extracted character shapes dataset is constructed through the manual extraction of every character from the set of Arabic words handwritten by multiple users.
- Provides a CNN based model to verify writers based on individual handwritten Arabic character shapes.
- Proposes a mechanism to identify the Arabic character shapes that are more effective for writer verification.
- Provides a comparative analysis of performance of writer verification based on isolated and extracted Arabic character shapes.

The rest of the paper is structured as follows. We start with an overview of the existing work being done related to writer identification in 'Related work'. Then, we describe our proposed writer verification approach in 'Proposed Approach'. It also describes the process used to develop our dataset. In 'Experimental Results and Discussion', we describe the experimental results and provide a discussion of these results. Finally, in 'Conclusions' conclusions are drawn. The list of abbreviations and symbols used in this paper are listed in Tables 1 and 2 respectively.

## RELATED WORK

This section present an extensive survey and review of the research literature pertaining to the writer identification and verification. Even though this domain encompasses various languages, we choose to focus on the Arabic language related research to limit our scope of work. We propose categorizing existing related literature into two broad categories which are based on the machine learning models being considered in the writer identification context. The first category (Conventional ML-based approach) is the one where the

**Table 1  List of abbreviations.**

| Abbreviation | Full Form |
| --- | --- |
| CNN | Convolutional Neural Network |
| AHAWP | Arabic Handwritten Alphabet, Words and Paragraphs Per User |
| ML | Machine Learning |
| DL | Deep Learning |
| GLRL | Grey Level Run Length |
| GLCM | Grey Level Co-occurrence Matrix |
| SSM | Spectral Statistical Measures |
| SVM | Support Vector Machines |
| GA | Genetic Algorithm |
| LDC | Linear Discriminant Classifier |
| WED | Weighted Euclidean Distance |
| SURF | Speed Up Robust Features |
| HMM | Hidden Markov Model |
| AHCD | Arabic Handwritten Character Dataset |
| SOM | Self Organising Maps |
| IAD | Isolated Alphabet Characters Dataset |
| EAD | Extracted Alphabet Characters Dataset |
| OVR | One Versus Rest |
| QUWI | Qatar University Writer Identification |
| KHATT | KFUPM Handwritten Arabic TexT |

**Table 2  List of symbols.**

| Symbol | Meaning |
| --- | --- |
| $a_i$ | Individual character |
| $w$ | Recovered Arabic text |
| $user_j$ | $j$th writer |
| $\beta_{user_j}(w)$ | Verification accuracy of recovered text ($w$) for $j$th writer |
| $\phi$ | Decision threshold |
| $\alpha_{user_j}(w)$ | Writer verification of recovered text ($w$) |
| $\beta(w)$ | Overall writer verification accuracy |
| $\Omega$ | Model accuracy |
| $\gamma_i$ | Recall |
| $\rho_i$ | Precision |
| $\delta_{ik}$ | Ratio of errors made by $i$th model against $k$th character |
| $\lambda_k$ | Average error of each character across all writers |

features from the handwritten text are extracted manually and then a machine learning (ML) classification algorithm is used for writer identification. In the second category (termed as a Deep Learning based approach) the features are extracted automatically using a Deep Learning ML model and simultaneously user identification is performed. Before we

proceed with reviewing both categories, we present a summary of two related important survey papers we came across during the literature search.

**Survey papers:** The first review paper presents a comprehensive summary (of about 50 research papers) of the research work related to Arabic writer identification. The authors classify the work based on text-dependent and text-independent studies and also based on writer identification *versus* verification. Also, a detailed summary of the various datasets related to this research field is presented. Finally, the papers are classified based on the feature set used in the classification process, the classifiers used and their relative performance measures in terms of accuracy (*Ahmed & Sulong, 2014*).

The second review paper provides a comprehensive review (of about 200 papers) in the domain of writer identification and verification (*Rehman, Naz & Razzak, 2019*). The authors provide a taxonomy of dataset, feature extraction methods and also provide a classification of research as conventional (manual feature extraction) and deep learning (automated feature extraction) methods. References have also been classified based on languages; namely, English, Arabic, Western and Other languages. Some of the inherent challenges have been discussed in this research domain with possible solutions and future directions. A more recent review paper which summarizes the recent work in this domain presents the various approaches, datasets being used, the challenges, and future directions of research related to Arabic handwritten character identification and verification (*Balaha, Ali & Badawy, 2021*).

**Conventional ML-based approaches**: In *Abdi & Khemakhem (2015)*, the authors propose, implement, and test a grapheme-based feature selection and classification process to identify and verify writers who have written Arabic documents. They claim that their approach uses a universal feature codebook which is not specific to any language), synthetic (not training-based) and model-driven (corpus independent). More specifically, they used a beta-elliptic model to synthesize their own grapheme-based codebook. Also, they use feature selection to reduce the size of this codebook to make their training and model efficient. The IFN/INIT dataset consisting of Arabic words written by 411 writers was used to train the model. Chi-square distance measure gave an identification accuracy of 96.35% and a verification error rate of 2.1%.

The work in *Djeddi & Souici Meslati (2010)* presents a texture-based approach where the writing style (which is visually distinctive) of each writer is considered as a texture. The texture definition is based on novel features extracted from Grey Level Run Length (GLRL) matrices. They also used the IFN/ENIT dataset to get the features from 2200 documents written in Arabic language. A comparison of the proposed approach is made with a popular Grey Level Co-occurrence Matrices (GLCM) technique and their approach gives better performance based on the fact that the GLRL matrices have more discriminatory features as compared to GLCM matrices. A chi-square distance measure for the proposed approach gave a classification accuracy of 96% taking into consideration the top-10 features of the GLRL matrices.

In *Al-Dmour & Zitar (2007)*, the authors have used manual extraction of features based on hybrid spectral-statistical measures (SSMs) of the Arabic handwriting texture. They have compared their approach with multi-channel Gabor filters and GLCM matrices for

feature extraction. Texture features included writing with a wide range of frequencies and orientations to make the features as generic as possible. The reduced feature set was obtained with a hybrid SVM-GA technique for making the model less complex. The writer identification results were obtained using four different classifiers; namely, Linear Discriminant Classifier (LDC), SVM, Weighted Euclidean Distance (WED), and k-NN with a maximum accuracy of 90%. In another paper, the authors propose and implement a writer identification technique based on extracting handwritten words that are characterized by two textural descriptors; namely, HOG and GLRL matrices. By fusing both similarity scores, they claim to have achieved a better writer identification. Their systems is tested on three datasets; namely, IFN/ENIT, KHATT and QUWI datasets which have handwritten documents from 411, 1,000 and 1,017 writers, respectively. The classification results that were achieved were 96.86%, 85.40% and 76.27% on the IFN/ENIT, KHATT and QUWI datasets, respectively (*Hannad et al., 2019*).

In *Maliki, Al-Jawad & Jassim (2017)*, the authors have proposed to generate features from sub-words rather than the whole words in the Arabic sentences or documents. This is a technique that falls in the category of text-dependent writer identification, where a dataset consists of 20 text samples from 95 writers. They identified 22 sub-words out of 49 which were contributing to a better performance in writer classification. The features were compared for similarity with two distance measure techniques; namely, Euclidean distance and Dynamic Time Warping. With this approach, a classification accuracy of 98% was achieved. One of the major drawback of this approach is that the experiments were conducted on an indigenous dataset which makes the comparison with other approaches not feasible. In a different technique of manual feature extraction, the authors proposed to avoid segmentation of words into sub-letters and used feature extraction techniques like Speed Up Robust Feature Transform (SURF) and k-NN to improve Arabic writer identification accuracy (*Abdul Hassan, Mahdi & Mohammed, 2019*). K-means algorithm was also utilized to identify and cluster similar features to improve the prediction process in terms of speed and accuracy. They have tested their approach on the IFN/ENIT dataset and have achieved a recognition rate of 96.6%.

In a different work (*Sheikh & Khotanlou, 2017*), the authors devised a Hidden Markov Model (HMM) based writer identification for the Persian (Farsi) writings. The HMM classifier was used to capture the angular characteristics of the written text. This resulted in a network chain of angular models leading to a comprehensive database for classification purposes. The same database was used during writer identification and have achieved an accuracy of about 60%, which is a bit on the lower side. This could be attributed to the complex structure of the Persian written text.

**Deep learning based approaches:** Most of the existing work using deep learning based techniques is used to automate the feature extraction phase but does not use deep learning based model as a classifier. For instance, the paper (*Rehman et al., 2019*) uses transfer learning to use the feature-set generated by the pre-trained AlexNet CNN model (5 Conv layers and 3 FC layers) on the ImageNet dataset. The authors have utilized these features on the QUWI dataset which consists of pages written in both English and Arabic languages and have performed both text-dependent (100,000 words) and text-independent (60,000

words) approaches for identifying the handwriting of 1,017 writers. The authors have performed segmentation of the document images where the text lines are extracted from the written paragraphs. The classification task is performed with the help of a multi-class SVM classifier. Data augmentation involves finding contours, sharpening and negative image generation of each word from the handwritten sentences. Performance results were obtained based on 80% of the data being used for training and 20% being used for testing purposes and have achieved an accuracy of 92.2% during the Arabic writer identification.

The work in *Kumar & Sharma (2020)* identifies the writer without the need for segmenting the lines or words from the Arabic handwritten document. It proposes and evaluates an approach that is based on CNN and weakly supervised region selection mechanism in the input image, which is the complete document. The basic idea behind avoiding segmentation of the document into lines and words is, to extract the features from a document for different depth levels using the CNN model, where a window of different sizes ($4 \times 4$, $8 \times 8$, and $16 \times 16$ grid cells) is applied. Then, features are selected from different cell regions of the document and a voting weight of each selected cell region is obtained. The class of the writer is then decided based on the combination of probability vectors of the selected cell with their weights. The authors have considered various languages and have used the IFN/ENIT dataset for Arabic writer identification. The proposed approach has achieved an accuracy of 98.24%.

An interesting work has also been presented in *Durou et al. (2019)* where a manual feature extraction based approach is compared with the automated feature extraction method. In the first category, SURF and SIFT methods were utilized during the feature extraction step and then the SVM and k-NN classifiers were used during writers classification. In the latter approach, the feature set was automatically generated with the help of CNN through the AlexNet model. The dataset used for this purpose was the ICFHR-2012 which has Arabic documents written by 200 writers. Results showed that the CNN based model outperformed the SVM and k-NN based approaches by at least 4–5% improvement in terms of classification accuracy.

Researchers have also used deep learning approaches for Arabic character recognition. Although this work is different from our domain, we present some of the recent work to highlight how CNN based approaches have been used on Arabic dataset. In a recent work (*Altwaijry & Al-Turaiki, 2021*), the authors have proposed a CNN-based Arabic Character Recognition. The CNN model was used to extract the features from the handwritten character and the softmax component of it was used to perform the classification task. The work used two databases for this purpose; namely, AHCD (Arabic Handwriting Dataset) and Hijja datasets. Based on the experimental results, they claimed a prediction accuracy of 97% for the AHCD and 88% for the Hijja dataset. This work focuses on the identification of Arabic characters rather than writer identification.

Similarly, the work presented in *El-Sawy, Loey & EL-Bakry (2017)* uses a CNN-based approach for recognition of handwritten Arabic letters. It does not address the problem of writer identification from Arabic character shapes but rather focuses on Arabic character recognition problem. The said approach in this paper is useful in providing insights on how to design the CNN model and how to extract letters from words or sentences in

this context. As a matter of fact, it uses 16,800 handwritten characters from 60 writers, where each writer wrote all the Arabic characters ten times on two separate forms. This data was fed to a CNN model which gave an accuracy of 94.9%. Recent works on applying CNN-based deep learning approaches have resulted in better accuracy (*Balaha et al., 2021a*; *Balaha et al., 2021b*). Deep learning has also been combined with other approaches such as Mathematical Morphology Operations (MMO) to provide for better character recognition (*Elkhayati, Elkettani & Mourchid, 2022*). Some of the authors have tested their approaches on already existing datasets, apart from providing newer datasets to aid in further research in this area (*Khosroshahi et al., 2021*).

**Specific application in forensics:** In order to put our proposed work in context, we performed a survey of literature in the domain of handwriting identification with a specific focus on its application in forensic analysis. The majority of the proposed approaches fell under the category of conventional ML-based. For instance, researchers in *Okawa & Yoshida (2017)* have used feature extraction based on pen pressure and shape on a dataset of handwritten text in Japanese Kanji characters (*Okawa & Yoshida, 2015*) collected with the participation of 54 users. An accuracy of 96% was achieved using the SVM classifier.

In a different work (*Parziale, Santoro & Marcelli, 2016*), the authors have proposed a process to generate statistical features (mean and variance) of English characters by measuring height, width and angles between the strokes of different characters. Other conventional ML modeling techniques which were considered were based on unsupervised approach of Self Organizing Maps (SOM) (*Schomaker, Franke & Bulacu, 2007*) and Neural Networks combined with Genetic Algorithm (*Pervouchine & Leedham, 2007*) for forensic analysis of English characters. Also, some of the other approaches were from non-ML domain such as Dynamic Time Warping applied to Allographs (*Niels, Vuurpijl & Schomaker, 2007*) and adopting feature-based codebooks generated by using Fourier and Wavelet transforms of the handwritten characters (*Kumar, Chanda & Sharma, 2014*). The research work in *He & Schomaker (2020)* is the closest relevant work to what we intend to achieve in this work and is based on combining the words-based CNN model with their fragments based models(called FragNet) and improving the classification accuracy to reach to 100% for the CERUG-EN dataset and 96.3%, 99.1% and 97.6%, respectively for IAM, CVL and Firemaker datasets. Their approach, however, did not consider the problem domain of writer verification of partially damaged documents.

Based on the summary of the literature survey presented in Table 3, it can be seen that there is quite little work in writer verification or identification in forensic analysis domain. To the best of our knowledge, there is no existing work on writer verification or identification of partially damaged handwritten Arabic documents. Additionally, existing writer identification and verification work has not used individual character shapes. This makes it difficult for us to compare our work with any of the existing work in a meaningful way. The main contribution of this paper is to propose and evaluate a CNN-based writer verification approach using handwritten Arabic character shapes and then use these trained models to provide writer verification of partially damaged handwritten Arabic documents during forensic analysis. The details of our proposed approach are described in the next

**Table 3  A comparative summary of most recent related work.**

| No. | Paper | Approach | Dataset | Results | Pros | Cons |
|---|---|---|---|---|---|---|
| 1 | *Sheikh & Khotanlou (2017)* | HMM | Persian Dataset | 60% | Catering to Persian language | Very low accuracy |
| 2 | *Hannad et al. (2019)* | HOG + GLRL | IFN, KHATT, QUWI | 96.86% | Dataset variation | Manual feature extraction |
| 3 | *Abdul Hassan, Mahdi & Mohammed (2019)* | SURF + k-NN | IFN/ENIT | 96.6% | Higher accuracy | Manual feature extraction |
| 4 | *Rehman, Naz & Razzak (2019)* | Review paper | Review paper | Review paper | 200 papers reviewed | Two years old |
| 5 | *He & Schomaker (2020)* | CNN (FragNet) | CERUG-EN | 100% | High accuracy | For English only |
| 6 | *Altwaijry & Al-Turaiki (2021)* | CNN | AHCD, Hijja | 97% | High accuracy | Character recognition only |
| 7 | *Balaha, Ali & Badawy (2021)* | Review paper | Review paper | Review paper | Most recent review | Less focus on Writer identification |
| 8 | *Balaha et al. (2021a)* | CNN (AHCR-DLS) | HMBD, CMATER, AIA9k | 100% | Good accuracy | No Writer identification |
| 9 | *Elkhayati, Elkettani & Mourchid (2022)* | CNN + MMO | IFN/ENIT | 97.35% | High accuracy | Manual segmentation |
| 10 | Proposed approach | CNN | Novel dataset | 96% | Character-based writer verification | Limited to Arabic characters |

section and the experimental results are presented in 'Experimental Results and Discussion'.

# PROPOSED APPROACH

As discussed in 'Introduction', our proposed approach addresses the following key concerns:

- Develop a CNN-based model to verify users based on their handwritten Arabic character shapes
- Use the CNN model trained on individual character shapes for writer verification of partially damaged Arabic documents (by extracting characters from user handwritten text)

In order to train the model on Arabic character shapes, we had to collect a dataset of user handwritten Arabic characters. However, the Arabic character writing style varies depending on whether the character is written as an isolated character (not part of a word) or as part of a word. For example, Fig. 3 shows the variations among the same characters written by two different users in isolation and as part of the word.

It can be seen that there are substantial variations in the same character written by the same user depending on whether it is written in isolation or as part of the word. We, therefore, had two possibilities for user handwritten character dataset collection:

| Users | Isolated Characters | Extracted Characters |
|-------|---------------------|----------------------|
| User01 | | |
| User12 | | |

**Figure 3** Isolated *vs.* extracted character shape variations.

- Each user writes all possible variants of Arabic characters (isolated characters)
- Each user writes certain Arabic text and then manually extract the Arabic character shapes from these words (extracted characters)

For a comparative analysis both datasets are collected and these datasets are referred to as: Isolated Alphabet-characters Dataset (IAD) and Extracted Alphabet-characters Dataset (EAD). The IAD dataset consists of Arabic alphabet characters written in isolation, while

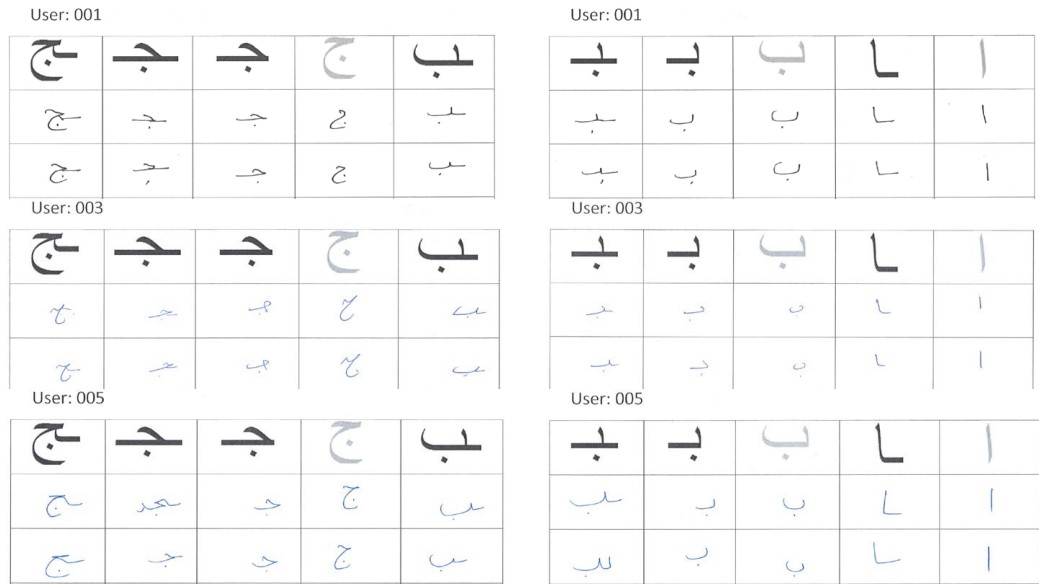

**Figure 4** Samples of handwritten isolated Arabic characters.

the EAD dataset consists of the alphabet characters extracted from user inscribed text. The details of these datasets are provided in the following subsections.

### Isolated alphabet characters dataset (IAD)

In the Arabic language, words use different forms of the same character depending on whether the character occurred at the beginning, middle, end, or in isolation (regular) as shown in Fig. 1. Further, Arabic alphabet characters can be classified into different groups depending on their similarity in writing style as shown in Fig. 1. Since the main objective of this research is to verify the writer of Arabic text recovered from partially damaged documents, which might contain any of these character shapes, we chose to collect all variants (begin, middle, end, and regular) of one character from each group. Thus, the dataset consists of 65 different variants of Arabic character shapes across 18 groups. The data was collected from eighty two (82) different Arabic writers in a classroom setting. Each user wrote each character shape ten times. This resulted in a dataset of 53, 199 isolated characters from 82 different users. Figure 4 shows samples collected from three different users. In real-world applications, users can use different writing instruments. We had advised the students to use a ballpoint pen but did not restrict them to any specific color.

### Extracted alphabet characters dataset (EAD)

The extracted characters dataset consisted of characters cropped from user handwritten Arabic text. The users were asked to write Arabic text (consisting of ten Arabic words). The set of words were selected such that they covered the entire set of Arabic alphabet characters (but not all character shape variations). Figure 5 shows a sample of user written text. The characters were extracted from these words manually, and a sample of extracted

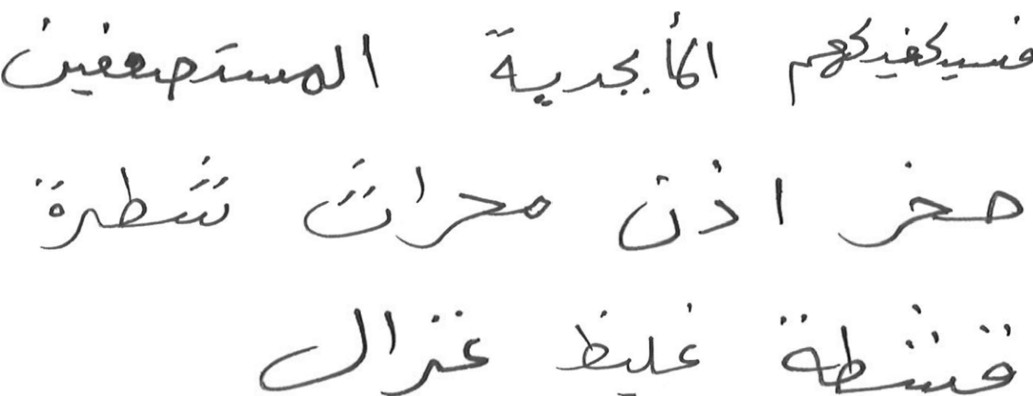

**Figure 5** Sample of user-written text.

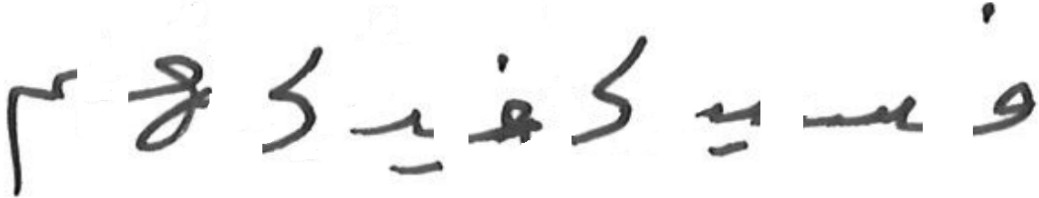

**Figure 6** Characters extracted from first user-written word.

characters from the first word is shown in Fig. 6. The complete dataset consists of 10, 780 extracted characters from different users.

Figure 7 shows the proposed methodology for writer verification using individual characters. The input characters dataset was pre-processed to remove any surrounding whitespace, converted to grayscale so that ink color does not become a distinguishing feature. The image size was condensed to $64 \times 64$ pixels for reducing the number of overall features. The pre-processed dataset was then split into training, validation, and test sets with a 60-20-20 ratio. It was also made sure that each character shape variant is present with the same ratio in each set. In order to implement user verification, we posed the problem as one *vs.* rest (OVR) classifier with one binary classifier per user. The reason for selecting the OVR approach was that, in our problem domain we focused on writer verification rather than writer identification. The OVR approach is a suitable approach in our problem domain (a typical forensic analysis case), where user verification is sought from a small group of suspects and generally does not need hundreds of users.

Each user would then be verified by using its own model. The dataset for each binary classifier represented a single classifier's training, validation, and test sets (with a 60-20-20 ratio) with two classes:

- target class (representing the target user of this classifier)
- rest class (represented the rest of the users).

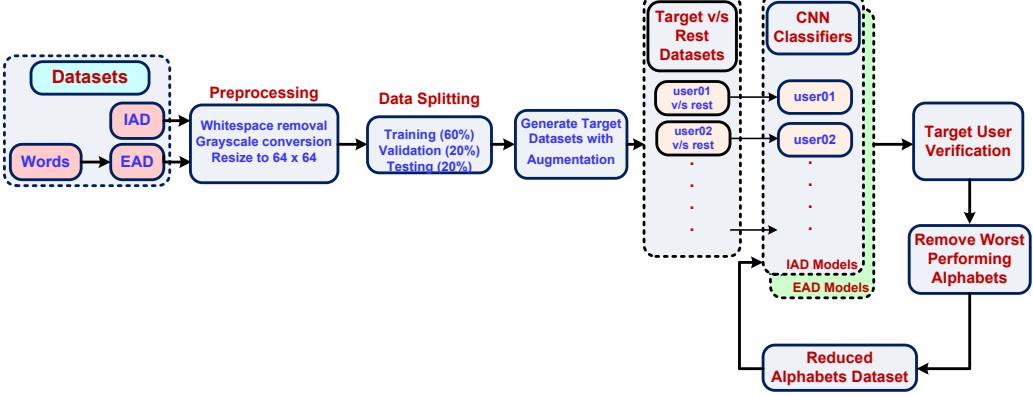

**Figure 7** Writer verification using Arabic characters.

Thus, the target class had fewer instances than the rest class. In order to balance the dataset, the target class data was augmented with a 5 percent random shift (left, right, up and down) along with a 10-degree random rotation. The CNN classifier was optimized using hyperparameter tuning to improve the validation accuracy. The trained models were then tested using each user's test set to determine test accuracy.

## Writer verification of partially damaged arabic documents

The individual writer verification models trained on isolated and extracted characters can then be used as components to verify authorship of damaged handwritten documents. It is a text-independent approach that can be used for any user written text. The approach works by extracting individual character ($a_i$) from each user written partially recovered text ($w$) where $w = a_1, a_2, \ldots, a_m$. Each $a_i \in w$ can then be used to verify the target user ($user_j$) using their corresponding character based model $f_{user_j}$ such that:

$$f_{user_j}(a_i) = \begin{cases} 1; & a_i \text{ is verified to be written by } user_j \\ 0; & a_i \text{ is not verified to be written by } user_j \end{cases} \tag{1}$$

The verification accuracy ($\beta_{user_j}(w)$) of the recovered text ($w$) for $user_j$ can thus be computed as:

$$\beta_{user_j}(w) = \frac{(\sum_i f_{user_j}(a_i))}{|w|} \tag{2}$$

We can define a threshold $\phi$ such that a document with text ($w$) is verified to be written by $user_j$, if ($\beta_{user_j}(w) >= \phi$). We will use the notation $\alpha_{user_j}(w)$ to refer to the writer verification of text $w$ for $user_j$:

$$\alpha_{user_j}(w) = \begin{cases} 1; & \beta_{user_j} >= \phi \\ 0; & \text{otherwise} \end{cases} \tag{3}$$

We will use the notation $\beta(w)$ to refer to average writer verification accuracy (across $n$ users) of user handwritten text $w$:

$$\beta(w) = \frac{(\sum_j \alpha_{user_j}(w))}{|n|} \tag{4}$$

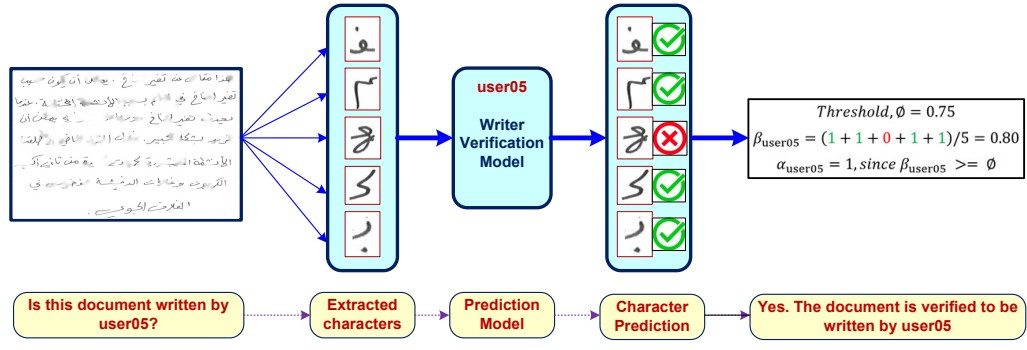

**Figure 8  A sample usage of the proposed writer verification approach.**

**Table 4  Experimental environment details.**

|  | Details |
|---|---|
| CPU | 3.70 GHz CPU with 6 cores |
| GPU Hardware | Nvidia GeForce GTX-1080 GPU with 2560 CUDA cores |
| GPU Memory | 32 GB |
| Programming language | Python |
| Library | Pandas, Keras and TensorFlow |
| Dataset | IAD and EAD |

In this paper we use $\phi = 0.75$ (*i.e.*, any handwritten text is verified to be written by $user_j$ if 75% of recovered characters from the document are verified to be written by $user_j$). A proper selection of $\phi$ value is application-dependent. Figure 8 shows how our proposed writer verification approach can be deployed to verify the authorship of any document.

## EXPERIMENTAL RESULTS AND DISCUSSION

The experiments were conducted on a GPU machine having 32 Gigabytes of memory, Nvidia GeForce GTX-1080 GPU with 2560 CUDA cores, and 3.70 GHz CPU with six cores. All the experiments were performed using the Python programming language with TensorFlow libraries. The experimental environment details are summarized in Table 4.

### Writer verification using isolated characters

Our initial analysis was conducted using the IAD dataset. The purpose of the analysis was to determine the efficacy of CNN-based approach to verify a user based on their handwritten isolated Arabic characters. We started with a CNN model with a single convolution and neural network layer. Figure 9A shows the model accuracy with this configuration. It can be seen that model is not able to learn well from the data and both training and validation accuracies are quite low (about 50%). We then incrementally added convolution layers and neural network layers with increased filter sizes until overfitting occurred; the middle plot in Fig. 9B shows the model with over-fitting. The dropout layers were then added to reduce overfitting resulting in a better configuration with model accuracies shown in Fig. 9C. The

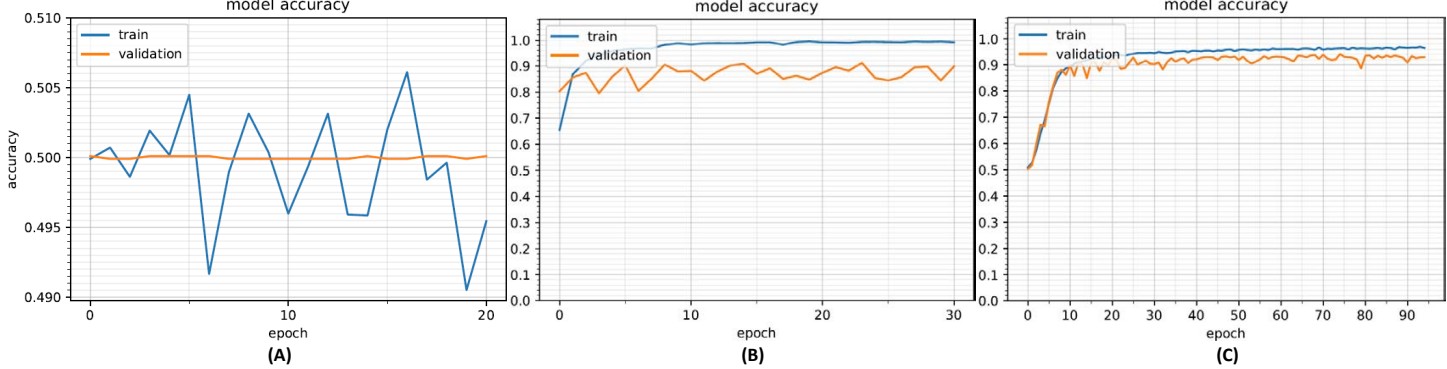

**Figure 9** Model accuracy with under-fit model (A), over-fit model (B) and optimized model (C).

**Table 5** The optimized CNN model used for training.

| Layer | Network layer | Output shape | Parameters |
|---|---|---|---|
| 1 | Convolution 1 | (62, 62, 128) | 1280 |
| 2 | Max Pooling 1 | (31, 31, 128) | 0 |
| 3 | Dropout 1 | (31, 31, 128) | 0 |
| 4 | Convolution 2 | (29, 29, 64) | 73792 |
| 5 | Max Pooling 2 | (14, 14, 64) | 0 |
| 6 | Dropout 2 | (14, 14, 64) | 0 |
| 7 | Convolution 3 | (12, 12, 64) | 36928 |
| 8 | Max Pooling 3 | (6, 6, 64) | 0 |
| 9 | Dropout 3 | (6, 6, 64) | 0 |
| 10 | Convolution 4 | (4, 4, 64) | 36928 |
| 11 | Max Pooling 4 | (2, 2, 64) | 0 |
| 12 | Flatten Layer | (256) | 0 |
| 13 | Dense Layer 1 | (128) | 32896 |
| 14 | Dropout 4 | (128) | 0 |
| 15 | Dense Layer 2 | (1) | 29 |

**Total parameters:** 181,953

optimized CNN model used for training purposes is shown in Table 5. The model takes as input $64 \times 64$ images and applies a convolutional layer with 128 filters (filter size $3 \times 3$). This is followed by an ELU activation layer to provide non-linearity and max pooling layer to extract prominent features and also reduce the features space. This was followed by three similar convolutional and max pooling layers. A dropout layer (probability = 0.5) was added after each max pooling layer to reduce overfitting.

The output of convolutional layers was 256 features that were then processed by a neural network hidden layer of 128 neurons followed by the output layer. Figure 10 shows the training, validation and test results for twenty randomly selected users' OVR models based on the IAD dataset. The color-coding scheme is used to highlight the minimum, maximum and variation in the results. The model accuracy is represented as $\Omega$. Therefore,

| userid | Model training | | Test_iad_all | | | test_iad_reduced | | | Test_ead | | |
|---|---|---|---|---|---|---|---|---|---|---|---|
| | $\Omega$ iad_training | $\Omega$ iad_validation | $\Omega$ iad_test | $\rho$ iad_test | $\Upsilon$ iad_test | $\Omega$ iad_reduced_test | $\rho$ iad_reduced_set | $\Upsilon$ iad_reduced_test | $\Omega$ ead_test | $\rho$ ead_test | $\Upsilon$ ead_test |
| user01 | 0.96 | 0.939 | 0.921 | 0.922 | 0.919 | 0.934 | 0.928 | 0.942 | 0.632 | 0.592 | 0.827 |
| user02 | 0.999 | 0.999 | 0.998 | 0.997 | 1 | 0.999 | 0.999 | 0.999 | 0.991 | 0.989 | 0.991 |
| user03 | 0.985 | 0.977 | 0.957 | 0.966 | 0.946 | 0.985 | 0.977 | 0.994 | 0.948 | 0.935 | 0.963 |
| user04 | 0.967 | 0.94 | 0.935 | 0.916 | 0.957 | 0.948 | 0.928 | 0.971 | 0.665 | 0.599 | 0.98 |
| user05 | 0.974 | 0.941 | 0.92 | 0.911 | 0.932 | 0.933 | 0.914 | 0.956 | 0.75 | 0.87 | 0.582 |
| user06 | 0.952 | 0.903 | 0.889 | 0.953 | 0.818 | 0.899 | 0.956 | 0.838 | 0.67 | 0.657 | 0.726 |
| user07 | 0.963 | 0.934 | 0.915 | 0.897 | 0.938 | 0.925 | 0.91 | 0.944 | 0.757 | 0.708 | 0.882 |
| user08 | 0.973 | 0.825 | 0.911 | 0.911 | 0.912 | 0.931 | 0.919 | 0.945 | 0.691 | 0.658 | 0.809 |
| user09 | 0.973 | 0.944 | 0.885 | 0.917 | 0.846 | 0.898 | 0.93 | 0.861 | 0.615 | 0.644 | 0.532 |
| user10 | 0.961 | 0.933 | 0.907 | 0.9 | 0.916 | 0.916 | 0.903 | 0.931 | 0.718 | 0.837 | 0.548 |
| user11 | 0.973 | 0.964 | 0.953 | 0.969 | 0.936 | 0.957 | 0.972 | 0.942 | 0.678 | 0.767 | 0.485 |
| user12 | 0.955 | 0.932 | 0.917 | 0.923 | 0.911 | 0.931 | 0.934 | 0.927 | 0.673 | 0.632 | 0.796 |
| user13 | 0.974 | 0.957 | 0.927 | 0.945 | 0.907 | 0.958 | 0.957 | 0.959 | 0.657 | 0.712 | 0.529 |
| user14 | 0.975 | 0.959 | 0.935 | 0.929 | 0.941 | 0.949 | 0.938 | 0.962 | 0.599 | 0.575 | 0.762 |
| user15 | 0.963 | 0.936 | 0.901 | 0.911 | 0.889 | 0.942 | 0.921 | 0.967 | 0.784 | 0.828 | 0.717 |
| user16 | 0.974 | 0.962 | 0.937 | 0.973 | 0.898 | 0.947 | 0.981 | 0.912 | 0.684 | 0.647 | 0.839 |
| user17 | 0.97 | 0.94 | 0.939 | 0.928 | 0.952 | 0.944 | 0.932 | 0.958 | 0.593 | 0.567 | 0.79 |
| user18 | 0.959 | 0.915 | 0.888 | 0.874 | 0.906 | 0.903 | 0.877 | 0.939 | 0.674 | 0.751 | 0.521 |
| user19 | 0.958 | 0.935 | 0.912 | 0.948 | 0.872 | 0.925 | 0.952 | 0.895 | 0.682 | 0.655 | 0.767 |
| user20 | 0.96 | 0.924 | 0.889 | 0.917 | 0.856 | 0.92 | 0.931 | 0.908 | 0.569 | 0.56 | 0.687 |
| Avg. | 0.97 | 0.94 | 0.92 | 0.93 | 0.91 | 0.94 | 0.94 | 0.94 | 0.70 | 0.71 | 0.74 |

**Figure 10   Model accuracy using the IAD dataset.**

$\Omega_{iad\_training}$ column shows the training accuracy and $\Omega_{iad\_validation}$ shows the validation accuracy during model training. It can be seen that average validation accuracy is 94% and the difference between training and validation accuracies is small. This indicates that the model has learnt quite well from the dataset.

The trained models were tested on the IAD test set. We represent recall and precision of $i$th model as $\gamma_i$ and $\rho_i$ respectively. Eqs. (5) and (6) show their calculations, where $\tau$ is the total number of correct target class predictions, $\xi$ is the total errors made to verify the target class, and $\eta$ is the total errors made by the model to incorrectly identify the other users as the target user. So, in essence, $\gamma$ shows the verification accuracy of the target user (*i.e.,* ratio of correct target class verification out of the target user written character shapes). Henceforth, we will use the term $\gamma$ as target user verification accuracy. The metric $\rho$ shows the ratio of correct target class verification out of all the target class predictions made by the model.

$$\gamma_i = \tau / (\tau + \xi) \tag{5}$$

$$\rho_i = \tau / (\tau + \eta) \tag{6}$$

The column "test_iad_all" in Fig. 10 shows the test accuracy ($\Omega_{test}$), precision ($\rho$) and target user verification accuracy ($\gamma_{iad\_test}$) for testing the model against all characters in the IAD test set. In this paper, we are mainly concerned with target user verification accuracy ($\gamma$). The average $\gamma_{iad\_test}$ is 91%, which indicates that the trained model works reasonably well on previously unseen isolated characters to verify the target user. Some users had a

| | | | | | | | | | | | |
|---|---|---|---|---|---|---|---|---|---|---|---|
| kaf_regular | كﺍ | 0.018 | seen_end | س | 0.053 | waw_end | و | 0.077 | **sad_middle** | ـصـ | 0.108 |
| feh_begin | ﻓ | 0.029 | beh_begin | ـﺑ | 0.054 | raa_end | ـر | 0.08 | **beh_regular** | ب | 0.109 |
| noon_end | ـﻥ | 0.032 | beh_middle | ـﺒـ | 0.057 | meem_end | ـﻢ | 0.082 | **lam_begin** | ل | 0.111 |
| yaa_middle | ـﻴـ | 0.033 | seen_middle | ـﺴـ | 0.057 | jeem_regular | ﺝ | 0.084 | **yaa_regular** | ي | 0.114 |
| heh_middle | ـﻬـ | 0.035 | sad_end | ـﺺ | 0.057 | meem_regular | م | 0.084 | **alif_end** | ـﺍ | 0.121 |
| qaf_middle | ـﻘـ | 0.037 | tah_regular | ﻁ | 0.057 | heh_end | ـﻪ | 0.084 | **waw_regular** | و | 0.121 |
| jeem_middle | ـﺠـ | 0.045 | sad_regular | ص | 0.058 | lam_middle | ـﻠـ | 0.085 | **kaf_begin** | ﻛ | 0.125 |
| seen_regular | س | 0.045 | jeem_begin | ﺟ | 0.059 | sad_begin | ﺻ | 0.087 | **seen_begin** | ﺳ | 0.128 |
| feh_middle | ـﻔـ | 0.045 | qaf_regular | ﻕ | 0.061 | kaf_middle | ـﻜـ | 0.087 | **heh_regular** | ه | 0.149 |
| beh_end | ـﺐ | 0.049 | ain_begin | ﻋ | 0.062 | meem_begin | ﻣ | 0.091 | **dal_regular** | د | 0.152 |
| tah_middle | ـﻄـ | 0.049 | ain_regular | ع | 0.062 | heh_begin | ﻫ | 0.091 | **raa_regular** | ر | 0.188 |
| feh_end | ـﻒ | 0.05 | noon_middle | ـﻨـ | 0.062 | noon_regular | ن | 0.096 | **lam_regular** | ل | 0.213 |
| qaf_end | ـﻖ | 0.05 | yaa_begin | ـﻴ | 0.063 | feh_regular | ف | 0.097 | **alif_hamza** | أ | 0.362 |
| yaa_end | ـﻯ | 0.05 | lam_alif | ﻻ | 0.063 | meem_middle | ـﻤـ | 0.097 | **alif_regular** | ﺍ | 0.39 |
| tah_end | ـﻂ | 0.051 | lam_end | ـﻞ | 0.067 | **ain_middle** | ـﻌـ | 0.1 | | | |
| kaf_end | ـﻚ | 0.051 | noon_begin | ـﻧ | 0.072 | **jeem_end** | ـﺞ | 0.104 | | | |
| qaf_begin | ﻗ | 0.052 | dal_end | ـﺪ | 0.074 | **ain_end** | ـﻊ | 0.107 | | | |

**Figure 11** **Average error ($\lambda_k$) of isolated characters across all users for the IAD dataset.**

low $\gamma_{iad\_test}$ values (*e.g.*, user06 has 82%) while a few others had a very high value of $\gamma_{iad\_test}$ (*e.g.*, 100% for user02). The very high validation and test accuracy attained by some users can be attributed to their unique writing styles.

In order to understand the reason behind the lower recall for some of the users, we had to look into the performance of the model on each user written character. We collected the ratio of verification errors made per character by each target user model. We represent the ratio of verification error made by $i$th target user model against $k$th character as $\delta_{ik}$ such that $\gamma_i = 1 - \sum_k \delta_{ik}$. Figure 11 shows the average error ($\lambda_k$) across all users for each character for the IAD dataset where $\lambda_k = \frac{\sum_i \delta_{ik}}{n}$ where $n$ is the total number of users. It can be seen that most of the characters got less than 10% error, but some characters (*e.g.*, alif_regular, lam_regular, *etc.*) had high errors. For example, alif_regular had a 40% average error. It can be attributed to the writing style of these characters, as alif_regular is written like a straight line and there would be quite less distinction in its writing style across users.

The average errors ($\lambda_k$) shown in Fig. 11 do not provide us with enough details on whether the errors were made by a single user as an outlier or spread across a large set of users. In order to understand the distributions of errors, we show the individual error values ($\delta_{ik}$) of two best, average and worst performing characters using heat map in Fig. 12. It can be seen that the best performing characters (kaf_regular and feh_begin) perform well across all users. The worst performing characters (alif_regular and alif_hamza) perform worst across the majority of the users.

Based on the above analysis, we can deduce that some character shapes have more distinguishing features while others have lesser distinguishing features for writer identification. Hence, it is better to ignore the worst performing character shapes for writer identification. We evaluated the model by eliminating the 25% worst performing character shapes (highlighted with bold font in Fig. 11). The results of $\Omega_{iad\_reduced\_test}$ and $\gamma_{iad\_reduced\_test}$ are shown in the "test_iad_reduced" column in Fig. 10. It can be seen that

| userid | kaf_regular ك | feh_begin فـ | qaf_end ـق | yaa_end ـي | alif_regular ا | alif_hamza أ |
|--------|------------|----------|--------|--------|--------------|------------|
| user01 | 0.053 | 0.079 | 0 | 0.105 | 0.553 | 0.447 |
| user02 | 0 | 0 | 0 | 0 | 0 | 0 |
| user03 | 0 | 0 | 0 | 0 | 0.595 | 0.816 |
| user04 | 0 | 0 | 0 | 0.289 | 0 | 0.132 |
| user05 | 0 | 0.026 | 0.132 | 0 | 0.447 | 0.342 |
| user06 | 0.026 | 0.053 | 0.053 | 0.132 | 0.711 | 0.184 |
| user07 | 0 | 0 | 0.026 | 0.026 | 0.026 | 0.605 |
| user08 | 0 | 0 | 0.158 | 0.026 | 0.553 | 0.5 |
| user09 | 0 | 0 | 0.184 | 0.026 | 0.026 | 0.526 |
| user10 | 0 | 0.053 | 0.132 | 0 | 0.447 | 0.158 |
| user11 | 0.026 | 0.026 | 0 | 0.026 | 0.079 | 0.553 |
| user12 | 0.053 | 0.053 | 0.053 | 0.053 | 0.289 | 0.132 |
| user13 | 0 | 0.079 | 0 | 0.105 | 0.421 | 0.605 |
| user14 | 0.026 | 0 | 0.026 | 0 | 0.5 | 0.447 |
| user15 | 0 | 0 | 0.026 | 0 | 0.947 | 0 |
| user16 | 0.053 | 0.053 | 0.026 | 0.026 | 0.526 | 0.026 |
| user17 | 0.053 | 0 | 0.053 | 0 | 0.053 | 0.474 |
| user18 | 0 | 0.053 | 0 | 0 | 0.5 | 0.316 |
| user19 | 0.079 | 0.079 | 0.079 | 0.053 | 0.763 | 0.526 |
| user20 | 0 | 0.026 | 0.053 | 0.132 | 0.368 | 0.447 |

**Figure 12** **Error ratio of best, average and worst performing characters across all users for the IAD dataset (darker color indicates higher error).**

the performance has improved for each user model with the reduced set of character shapes. The average model performance improved to 93.75% from 91.25%. The model trained on the IAD dataset performed quite well on the test set of isolated characters. However, in practice, we need to verify the writer based on written text rather than just the isolated characters. Therefore, we evaluated model performance on characters extracted from user written text by testing it against the test set of the EAD dataset. The column "test_ead" in Fig. 10 shows the $\Omega_{ead\_test}$ and $\gamma_{ead\_test}$ values for the EAD test set. The average $\gamma_{ead\_test}$ was a meager 74% and six out of twenty users had $\gamma_{ead\_test}$ values close to 50%. This means that model trained on the IAD dataset does not perform well on characters extracted from the text. As anticipated, the isolated characters are quite different from extracted character and therefore cannot be used as a reliable model to predict user written text.

## Writer verification using extracted characters

As seen in the previous experiments, the models trained on isolated characters cannot be used reliably to identify user written text (*i.e.,* characters extracted from the text). Therefore, we evaluated a CNN based OVR model that was trained using the EAD dataset. The results obtained for twenty randomly selected users' models are shown in Fig. 13. The average training and validation accuracies ($\Omega_{ead\_training}$ and $\Omega_{ead\_validation}$) of these models was 97.5% and 92% respectively. This shows that the models learned well on training data. Test

| userid | model training | | test_ead_all | | test_ead_reduced | |
|---|---|---|---|---|---|---|
| | $\Omega_{ead\_training}$ | $\Omega_{ead\_validation}$ | $\Omega_{ead\_test}$ | $\Upsilon_{ead\_test}$ | $\Omega_{ead\_reduced\_test}$ | $\Upsilon_{ead\_reduced\_test}$ |
| user01 | 0.974 | 0.9 | 0.888 | 0.845 | 0.895 | 0.854 |
| user02 | 0.998 | 0.995 | 0.987 | 0.986 | 0.984 | 0.981 |
| user03 | 0.992 | 0.983 | 0.979 | 0.998 | 0.984 | 1 |
| user04 | 0.97 | 0.913 | 0.893 | 0.925 | 0.909 | 0.948 |
| user05 | 0.976 | 0.931 | 0.878 | 0.857 | 0.902 | 0.89 |
| user06 | 0.947 | 0.804 | 0.72 | 0.529 | 0.728 | 0.544 |
| user07 | 0.977 | 0.959 | 0.924 | 0.921 | 0.931 | 0.935 |
| user08 | 0.961 | 0.91 | 0.847 | 0.825 | 0.882 | 0.876 |
| user09 | 0.961 | 0.879 | 0.866 | 0.843 | 0.883 | 0.864 |
| user10 | 0.966 | 0.875 | 0.848 | 0.775 | 0.881 | 0.839 |
| user11 | 0.988 | 0.976 | 0.973 | 0.971 | 0.976 | 0.977 |
| user12 | 0.973 | 0.892 | 0.858 | 0.801 | 0.873 | 0.827 |
| user13 | 0.969 | 0.912 | 0.886 | 0.865 | 0.903 | 0.898 |
| user14 | 0.992 | 0.976 | 0.945 | 0.938 | 0.944 | 0.926 |
| user15 | 0.981 | 0.938 | 0.933 | 0.933 | 0.941 | 0.943 |
| user16 | 0.98 | 0.952 | 0.932 | 0.902 | 0.956 | 0.943 |
| user17 | 0.972 | 0.904 | 0.833 | 0.714 | 0.846 | 0.736 |
| user18 | 0.969 | 0.912 | 0.916 | 0.895 | 0.931 | 0.926 |
| user19 | 0.972 | 0.899 | 0.862 | 0.758 | 0.876 | 0.787 |
| user20 | 0.976 | 0.876 | 0.869 | 0.787 | 0.863 | 0.775 |
| **Average** | **0.97** | **0.92** | **0.89** | **0.85** | **0.90** | **0.87** |

**Figure 13** Model accuracies using the EAD dataset.

accuracy ($\Omega_{ead\_test}$) was also quite close to validation accuracy (89.2%). However, target user verification accuracy ($\gamma_{ead\_test}$) was close to 85% which is lower than the target user verification accuracy of isolated characters ($\gamma_{iad\_all} = 91.3\%$). Upon further investigation, it was found that this can be attributed to the presence of large variations within the extracted character shapes for the same user. In contrast, the isolated character shapes of the same user did not have such a large variation. When users are writing words in a flow, the shape of the same character changes across words. The shape of the character also varies depending upon how the writer joins it with the neighboring characters. To illustrate these variations in the characters written by the same user, samples of two different character shapes (ain_middle and yaa_middle) written by same user (*user05*) are shown in Fig. 14.

It can also be noticed that some user verification models did not perform well, for example, *user06* had target user verification accuracy of only 52.9%. On closer inspection, it was found that the model performed badly with more than 80% error on few characters (jeem_middle, feh_middle, ain_middle, noon_end, alif_hamza, lam_alif). For example, the average error on character ''jeem_middle'' from other user models was 13.1%, but the *user06* model had an error of 94.7%. Similarly, character ''ain_middle'' had 97.4% error

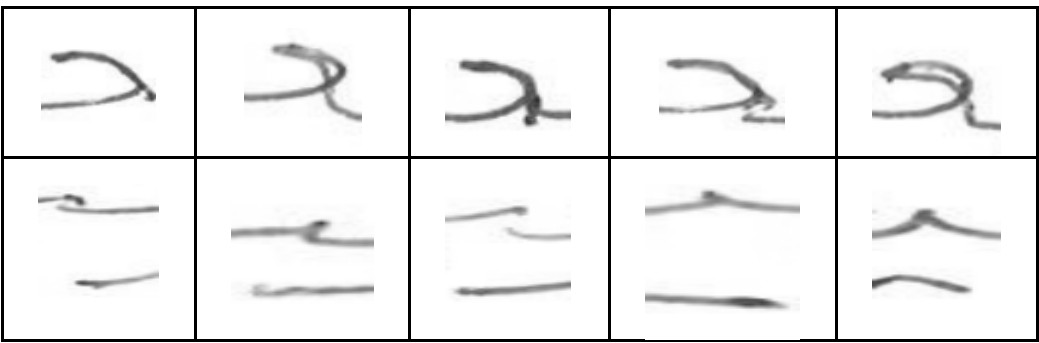

**Figure 14** Variations in extracted ain_middle (top) and yaa_middle (bottom) characters written by same user (*user05*).

| | | | | | | | | | | | | |
|---|---|---|---|---|---|---|---|---|---|---|---|---|
| lam_begin | ل | 0.045 | sad_begin | ص | 0.097 | feh_middle | ف | 0.16 | **haa_middle** | ح | 0.174 |
| khah_middle | خ | 0.047 | feh_begin | ف | 0.111 | zay_end | ز | 0.161 | **teh_middle** | ت | 0.176 |
| qaf_begin | ق | 0.053 | theh_regular | ث | 0.124 | alif_regular | ا | 0.162 | **lam_middle** | ل | 0.18 |
| lam_alif | لأ | 0.068 | noon_regular | ن | 0.134 | ghain_begin | غ | 0.162 | **yaa_begin** | ي | 0.181 |
| dad_middle | ض | 0.071 | sheen_begin | ش | 0.14 | ain_middle | ع | 0.163 | **raa_end** | ر | 0.196 |
| tah_middle | ط | 0.073 | heh_middle | ﻬ | 0.141 | thal_regular | ذ | 0.168 | **meem_middle** | م | 0.256 |
| sheen_middle | ش | 0.079 | teh_closed | ة | 0.141 | **dal_end** | ـد | 0.171 | **heh_regular** | ه | 0.256 |
| kaf_middle | ك | 0.082 | meem_begin | م | 0.15 | **meem_end** | م | 0.171 | **alif_hamza** | أ | 0.265 |
| thah_end | ظ | 0.096 | lam_regular | ل | 0.158 | **seen_middle** | س | 0.172 | **noon_end** | ن | 0.272 |
| beh_begin | ب | 0.097 | yaa_middle | ﻴ | 0.159 | **jeem_middle** | ﺟ | 0.173 | | | |

**Figure 15** Average error ($\lambda_k$) of extracted characters across all users for the EAD dataset.

for *user06* model while the average error for other users is only 12%. This large error is due to the resemblance of these characters with other users' handwritten characters.

We used the same methodology as described in 'Writer verification using isolated characters' to identify the performance of individual characters. The average errors ($\lambda_k$) are shown in Fig. 15 and the 33% worst performing characters (having an average error larger than 17%) are highlighted in bold font. We re-evaluated the model using the reduced set of characters (*i.e.,* using characters which are not highlighted with bold font in Fig. 15). The results of $\Omega_{ead\_reduced\_test}$ and $\gamma_{ead\_reduced\_test}$ are shown in the ''test_ead_reduced'' column in Fig. 13. It can be seen that the performance has improved for each user model with the reduced set of characters. The average model performance improved to 87.3% from 85.3%.

It can also be observed in Fig. 15 that the similar shaped characters in EAD dataset have large variation in their errors. For example, we have noticed the disparities in average error of following similar shaped characters:

- yaa_begin (18.1%) and beh_begin (9.7%)

- Haa_middle (17.4%), Khah_middle (4.7%)
- Feh_begin (11.1%), Qaf_begin (5.3%)
- Teh_middle (17.6%), Theh_regular (12.4%)
- Noon_end (27.2%), Noon_regular(13.4%)

Upon further investigations, we discovered that sometimes a character shape written by a user matches with a different character shape of another user (when character shapes are similar). For example, we noticed that user09 did not put dots in the right place for yaa_begin. This resulted in a large error because the model took it as beh_begin of user07. Due to this reason, yaa_begin has larger error (18.1%) than beh_begin (9.7%). Similar observations were made for other similar shaped character in EAD. Noon_end has the highest error because of its high similarity in writing style across several users and also similarity with Noon_regular written by other users. Based on the above observations, it seems desirable to use only a single character from the group of similar character shapes during extraction process to reduce the overall chances of verification errors.

These issues were less prominent in IAD because of character shapes written in isolation being more consistent, accurate (*e.g.*, no dots related issues) and our choice of using a single character shape from within the group of similar shaped characters (as shown in Fig. 1). This resulted in having lesser variations in character shapes written by the same user and also lesser chances of errors caused by similar shaped characters. For instance, for IAD dataset the errors for Yaa-begin, Beh-begin, Noon-begin are 6.3%, 5.4% and 7.2% respectively, which has relatively smaller differences in errors.

## Using character based models for writer verification of partially damaged documents

In this section, we describe the evaluation of the character based models (as mentioned in 'Writer verification using extracted characters') for writer verification of partially damaged documents. Each character extracted from the recovered text $w$ written by $user_j$ was checked using the EAD model ($f_{user_j}(a_i)$). The writer verification accuracy of recovered document ($\beta_{user_j}(w)$) for twenty randomly selected users is shown in Fig. 16. The overall writer verification accuracy of the damaged documents ($\beta(w)$) is also shown at the end of the table.

These experiments were conducted using the Arabic text shown in Fig. 5. Experiments were conducted with varying percentages of recovered characters from the text (80% to 10% document recovery). For each recovered document, experiments were conducted varying the percentage of good performing characters in the recovered characters (10% to 90%). The set of good performing characters was based on Fig. 15 with an average error of less than 17%. The rest of the characters (shown in bold font in Fig. 15) were considered bad performing characters. Each experiment was repeated ten times with a random selection of good and bad performing characters from within the text.

It can be seen that overall user verification accuracy is highly dependent on the quality of characters recovered and it varies between 85% (with good performing character shapes =10%) to 95% (with good performing character shapes =90%). We get similar results when the percentage of a recovered document is smaller (say 10% document recovered).

| userid | Percentage of good performance characters within recovered document | | | | | | | | | | | | | | | | | | | |
|---|---|---|---|---|---|---|---|---|---|---|---|---|---|---|---|---|---|---|---|---|
| | 10% | 20% | 50% | 75% | 90% | 10% | 20% | 50% | 75% | 90% | 10% | 20% | 50% | 75% | 90% | 10% | 20% | 50% | 75% | 90% |
| user01 | 0.77 | 0.8 | 0.78 | 0.86 | 0.87 | 0.8 | 0.79 | 0.83 | 0.88 | 0.88 | 0.74 | 0.78 | 0.84 | 0.85 | 0.86 | 0.75 | 0.82 | 0.88 | 0.87 | 0.9 |
| user02 | 0.93 | 0.93 | 0.97 | 0.98 | 0.99 | 0.92 | 0.93 | 0.96 | 0.99 | 1 | 0.94 | 0.96 | 0.95 | 1 | 0.98 | 0.95 | 0.95 | 0.98 | 0.98 | 1 |
| user03 | 0.96 | 0.95 | 0.96 | 0.98 | 0.99 | 0.93 | 0.91 | 0.98 | 0.99 | 0.99 | 0.93 | 0.95 | 0.96 | 0.99 | 1 | 0.93 | 0.95 | 0.92 | 0.98 | 1 |
| user04 | 0.94 | 0.93 | 0.95 | 0.96 | 0.97 | 0.93 | 0.9 | 0.93 | 0.93 | 0.95 | 0.95 | 0.9 | 0.97 | 0.96 | 0.98 | 0.95 | 0.97 | 0.97 | 0.98 | 0.97 |
| user05 | 0.76 | 0.78 | 0.82 | 0.85 | 0.9 | 0.8 | 0.77 | 0.79 | 0.85 | 0.9 | 0.89 | 0.75 | 0.86 | 0.86 | 0.92 | 0.85 | 0.87 | 0.9 | 0.8 | 0.9 |
| user06 | 0.49 | 0.49 | 0.57 | 0.57 | 0.68 | 0.47 | 0.53 | 0.57 | 0.64 | 0.65 | 0.54 | 0.57 | 0.64 | 0.72 | 0.69 | 0.52 | 0.43 | 0.6 | 0.55 | 0.53 |
| user07 | 0.99 | 0.99 | 0.98 | 0.98 | 0.97 | 0.99 | 1 | 0.98 | 0.95 | 0.98 | 0.99 | 0.98 | 0.98 | 0.97 | 0.96 | 1 | 1 | 0.95 | 0.98 | 0.98 |
| user08 | 0.88 | 0.86 | 0.89 | 0.91 | 0.95 | 0.87 | 0.87 | 0.9 | 0.94 | 0.95 | 0.85 | 0.91 | 0.92 | 0.95 | 0.95 | 0.85 | 0.93 | 0.9 | 0.88 | 0.95 |
| user09 | 0.87 | 0.87 | 0.86 | 0.88 | 0.86 | 0.89 | 0.82 | 0.85 | 0.87 | 0.88 | 0.85 | 0.85 | 0.85 | 0.92 | 0.84 | 0.9 | 0.92 | 0.85 | 0.92 | 0.88 |
| user10 | 0.77 | 0.75 | 0.86 | 0.92 | 0.96 | 0.72 | 0.76 | 0.84 | 0.94 | 0.97 | 0.7 | 0.76 | 0.92 | 0.94 | 0.96 | 0.77 | 0.77 | 0.85 | 0.92 | 1 |
| user11 | 0.99 | 0.99 | 0.95 | 0.95 | 0.93 | 0.99 | 0.99 | 0.96 | 0.94 | 0.91 | 0.97 | 0.99 | 0.97 | 0.95 | 0.94 | 0.97 | 1 | 0.9 | 0.93 | 0.85 |
| user12 | 0.63 | 0.65 | 0.73 | 0.7 | 0.81 | 0.66 | 0.63 | 0.72 | 0.72 | 0.81 | 0.69 | 0.59 | 0.71 | 0.75 | 0.78 | 0.67 | 0.67 | 0.58 | 0.75 | 0.8 |
| user13 | 0.88 | 0.86 | 0.89 | 0.9 | 0.92 | 0.82 | 0.88 | 0.89 | 0.89 | 0.92 | 0.91 | 0.91 | 0.89 | 0.95 | 0.92 | 0.9 | 0.87 | 0.93 | 0.92 | 0.9 |
| user14 | 0.85 | 0.88 | 0.92 | 0.94 | 0.99 | 0.85 | 0.87 | 0.94 | 0.97 | 0.99 | 0.84 | 0.86 | 0.94 | 0.98 | 1 | 0.88 | 0.83 | 0.93 | 0.98 | 1 |
| user15 | 0.84 | 0.86 | 0.93 | 0.97 | 0.98 | 0.9 | 0.9 | 0.88 | 0.97 | 0.99 | 0.9 | 0.91 | 0.93 | 0.97 | 0.98 | 0.87 | 0.83 | 0.93 | 0.97 | 1 |
| user16 | 0.8 | 0.8 | 0.88 | 0.94 | 0.98 | 0.81 | 0.78 | 0.89 | 0.93 | 0.99 | 0.77 | 0.83 | 0.92 | 0.96 | 0.98 | 0.82 | 0.82 | 0.92 | 0.98 | 1 |
| user17 | 0.72 | 0.68 | 0.73 | 0.79 | 0.78 | 0.7 | 0.66 | 0.69 | 0.78 | 0.76 | 0.7 | 0.68 | 0.74 | 0.74 | 0.79 | 0.7 | 0.62 | 0.9 | 0.73 | 0.65 |
| user18 | 0.79 | 0.79 | 0.88 | 0.95 | 0.98 | 0.81 | 0.79 | 0.87 | 0.94 | 0.99 | 0.74 | 0.83 | 0.93 | 0.97 | 0.97 | 0.85 | 0.9 | 0.87 | 0.95 | 1 |
| user19 | 0.78 | 0.74 | 0.83 | 0.84 | 0.88 | 0.79 | 0.8 | 0.8 | 0.84 | 0.9 | 0.82 | 0.83 | 0.86 | 0.86 | 0.89 | 0.7 | 0.8 | 0.78 | 0.85 | 0.92 |
| user20 | 0.8 | 0.75 | 0.82 | 0.82 | 0.86 | 0.76 | 0.75 | 0.81 | 0.83 | 0.86 | 0.8 | 0.82 | 0.8 | 0.81 | 0.82 | 0.9 | 0.83 | 0.82 | 0.87 | 0.85 |
| $\beta(w)(\emptyset = 0.75)$ | 0.85 | 0.8 | 0.85 | 0.9 | **0.95** | 0.8 | 0.85 | 0.85 | 0.9 | **0.95** | 0.7 | 0.85 | 0.85 | 0.9 | **0.95** | 0.8 | 0.85 | 0.9 | 0.9 | **0.9** |
| | 80% document recovered | | | | | 50% document recovered | | | | | 20% document recovered | | | | | 10% document recovered | | | | |

**Figure 16** Writer verification accuracy ($\beta_{user_j}(w)$) of partially damaged documents with varying percentage of good performing character shapes in the recovered text.

During forensic analysis, writer verification based on documents with higher recovered characters and higher overall accuracy would be preferred over writer verification based on lower recovery and lower overall accuracy. In some problem domains, forensic experts might be willing to trade off document recovery and only seek high-performance characters to increase the writer verification accuracy but in some other problem domains, experts might want to figure out verification accuracy based on the completely recovered document, irrespective of character shape quality.

## CONCLUSIONS

This article provides a mechanism for writer verification of partially damaged handwritten documents (*e.g.*, during forensic analysis) where complete text is unavailable, but certain characters can still be extracted. The article proposes an individual character based approach for text-independent writer verification. The writer verification models based on isolated and extracted character shapes were developed using CNN. The article shows that writer verification based on individual isolated characters can be improved from 91% to 94% by eliminating the characters which do not provide any useful information to verify the writer. The article shows a similar writer verification approach based on the characters extracted from user-written text. The writer verification accuracy based on extracted characters

improves from 85% to 87% on a reduced set of good performing characters. The model performance on extracted characters is lower than on isolated characters because of the inconsistencies in user writing of characters as part of a word (depending on where the characters occur in the word). It was also considered desirable to extract only a single character from the group of similar shaped alphabet characters during extraction process, to reduce the chances of verification errors due to similarity of character shapes. Overall, it is shown that a writer verification accuracy between 80% to 95% can be attained for partially damaged documents of several degrees depending on the percentage of good performing character shapes extracted from the recovered document.

### Limitations and future work

The work has few limitations and therefore can be extended in multiple ways. Although our EAD dataset was based on a carefully selected group of Arabic words that covered all Arabic alphabet characters, it did not include all character shapes (*i.e.,* begin, middle, end and regular variants). A more comprehensive extracted alphabet characters dataset should be considered to include all character shapes. Additionally, writer identification (rather than just verification) of partially damaged handwritten documents would be a more challenging task. Also, the extraction of character shapes from handwritten text is a manual and tedious process which limits the scalability of our approach. This can be overcome with some other techniques such as automatically identifying intact parts of the document using object detection techniques and then using existing CNN based approaches on the recovered documents. It would also be interesting to evaluate the impact of "transfer learning" on the accuracy of the developed models.

### Funding
The authors received no funding for this work.

### Competing Interests
The authors declare there are no competing interests.

### Author Contributions
- Majid A. Khan conceived and designed the experiments, performed the experiments, analyzed the data, performed the computation work, prepared figures and/or tables, authored or reviewed drafts of the paper, and approved the final draft.
- Nazeeruddin Mohammad, Ghassen Ben Brahim and Abul Bashar conceived and designed the experiments, analyzed the data, prepared figures and/or tables, authored or reviewed drafts of the paper, and approved the final draft.
- Ghazanfar Latif conceived and designed the experiments, analyzed the data, authored or reviewed drafts of the paper, and approved the final draft.

### Data Availability
Raw data and code are available at Figshare:

Khan, Majid (2021): ReadMe.txt. figshare. Software. https://doi.org/10.6084/m9.figshare.17068562.v1.

Khan, Majid (2021): ReadMe.txt. figshare. Software. https://doi.org/10.6084/m9.figshare.17068562.v1.

Khan, Majid (2021): ReadMe.txt. figshare. Software. https://doi.org/10.6084/m9.figshare.17068562.v1.

## Supplemental Information

Supplemental information for this article can be found online at http://dx.doi.org/10.7717/peerj-cs.955#supplemental-information.

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
