# Peer review of "Writer verification of partially damaged handwritten Arabic documents based on individual character shapes"

_PeerJ Computer Science, doi:10.7717/peerj-cs.955_

## Round 0.1 · original submission · Major Revisions

I recommend a major revision before further consideration. Please provide a detailed response letter together with your revised paper. Thanks.

·

Basic reporting

[ABSTRACT] The abstract should reflect the contributions of the manuscript. I suggest rewriting it.
[INTRODUCTION] The authors should provide a clear problem definition and contributions in the introduction section.
[RELATED WORK] Where are the related studies? They should be declared in a separate section.
[RELATED WORK] A table of comparisons should be added at the end of the related studies section to praise the pros. and cons. of them. The year column should be added and they should be ordered by it.
[EQUATIONS] The authors should follow the journal authors’ guidance in writing the equations, symbols, and variables. Please, refer to the authors guidelines on the journal official website.
[EQUATIONS] Where are the equations of the used metrics?
[DATASETS] Samples from the used dataset should be added and annotated.
[METHODOLOGY] The suggested approach is not clearly discussed. More scientific details should be added.
[METHODOLOGY] What are the used equations in the suggested approach? In other words, how the suggested approach is derived?
[METHODOLOGY] Where is the overall pseudocode? Flowchart? of the suggested approach?
[ABBREVIATIONS] The authors should add a table of abbreviations in the revised manuscript.
[SYMBOLS] The authors should add a table of symbols in the revised manuscript.
[REFERENCES] There are no citations for many sentences in the manuscript. Why? Please check.
[REFERENCES] The references should be written in the same style following the journal authors’ guidance.
[REFERENCES] Recent citations from 2021 should be added to the manuscript. Only one is found.
[PROOFING] The authors should get editing help from someone with full professional proficiency in English.
[PROOFING] The manuscript should be checked again to fix any typos such as missing spaces and commas.
[CONSISTENCY] The manuscript structure is too short. It must be elaborated in their applied technology as should support more rigorous technical aspects.
[CONSISTENCY] Some paragraphs are wrapped in more than 10 lines. They should be split concisely.

Experimental design

[EXPERIMENTS] The experimental configurations (i.e., settings) should be declared and added to a table.
[EXPERIMENTS] The working environment (i.e., software and hardware) should be declared and added to a table.
[EXPERIMENTS] What are the criteria for selecting the experimental configurations?
[EXPERIMENTS] More experiments should be conducted using different configurations.
[EXPERIMENTS] Where is the tabular representation of the reported results?
[EXPERIMENTS] The figures in the experiments section should be gridded. For example, Figure 8.
[EXPERIMENTS] Why did not the authors compare their approach with others in a table?
[EXPERIMENTS] Why did not the authors compare their approach with another approach to compare the suggested approach efficiency and applicability?
[EXPERIMENTS] Can the authors draw the area under the curve (AUC)?
[EXPERIMENTS] Why did not the authors calculate other performance metrics such as specificity and f1-score?
[EXPERIMENTS] Where is the detailed and statistical discussion of the reported results?
[EXPERIMENTS] More experiments should be conducted using a different dataset to prove the generalization.

Validity of the findings

[NOVELTY] What is the novelty of the suggested approach?
[CONCLUSIONS] The conclusions in this manuscript are primitive. Please, write your conclusions.
Why did not the authors use transfer learning? More experiments should be conducted using it.
Why the authors added Figure 8?
In Table 1: Why the authors selected that model? What is the used criteria?
[LIMITATIONS] What are the limitations of the current study? It should be added in a separate section.

Additional comments

The manuscript presented “a verification of partially damaged handwritten Arabic documents approach”. However, the major and critical weak points are that:
(1) Their proposed work discussion is weak distributed to be described or analyzed.
(2) The novelty is not guaranteed.
(3) Their work is not compared with state-of-the-art approaches nor related studies.
(4) Their experiments leak from the descriptive and statistical analysis.

Reviewer 2 ·

Basic reporting

The article proposes an approach for offline text independent writer verification of partially damaged handwritten Arabic documents, based on individual characters. The extraction of "alphabets" from handwritten text is a manual process. The idea is excellent, however, my main reservation is on the procedure, dataset used and results reached. The dataset collection itself is excellent, however, I suggest to expand it and make it based on all Arabic Character Shapes.

Although the write up is professional there are some inconsistencies in some terms. Example of "handwritten" is spelled in 3 different ways: Two words (hand written) like in the abstract and line 88, (hand-written) as in line 254 and one word (handwritten) as in line 252 etc.. The authors use the term alphabets, letters, characters for Arabic. The most accepted term in the literature is character shape. Arabic characters take different shapes depending on their position in a word.

Better to write the Arabic character shape in Arabic as in line 47 it helps the reader to see the character shape in question. The name of the character shape is not enough.

Experimental design

The authors claim they are using all Arabic Alphabets. Their Extracted Alphabets Dataset (EAD) is collected from users using text in Figure 5. The text uses all Arabic Alphabet, however, it does not use all Arabic Character Shapes. Only about a quarter of the shapes are present in the text used. I suggest to expand the collection for other shapes as some missing shapes have very discriminatory features that could improve the writer verification and will be useful for an extension to writer identification. In this way I claim your reduced model will perform even better.

I don't see the advantage of having IAD and EAD separate. IAD is only one character shape among the 4 possible shapes that an Arabic alphabet can take. The difference is only in the segmentation of Words into characters. If the extraction is manual where is the advantage, please clarify.


Not clear:
* our domain of work .. does not address the problem of writer identification from the Arabic letters. Our main focus is on how to identify the writer based on the "given" handwritten Arabic letters.

* Comparison of isolated and extracted Arabic alphabet based approaches for writer verification.
* Isolated Alphabet:
Samples of user written isolated alphabets in Figure 4, is the middle column as I understand it. What is isolated alphabets in Figure 3, these are not the isolated shapes. Both columns are for middle shape or begin shape characters.

Validity of the findings

I have a serious reservation about the findings.
Why the same character body shape is giving different average error or average accuracy like in Figure 9 or Figure 11.
Examples:
Yaa-begin, Beh-begin, Noon-begin
Teh, Theh, Haa, Khaa
Feh-begin, Qaf-begin
Feh-middle, Qaf-middle
etc..

Additional comments

I encourage you to address the comments and reservation above and resubmit for eventual publication of your work.

---

## Round 0.2 · Minor Revisions

A further revision is needed to improve the presentation and address the reviewer's concerns.

·

Basic reporting

Thanks to the authors for updating the manuscript. After checking their responses to my comments, I can declare that they have made a suitable major revision.

Experimental design

Thanks to the authors for updating the manuscript. After checking their responses to my comments, I can declare that they have made a suitable major revision.

Validity of the findings

Thanks to the authors for updating the manuscript. After checking their responses to my comments, I can declare that they have made a suitable major revision.

Additional comments

Journal: PeerJ Computer Science
Manuscript Title: Writer Verification of Partially Damaged Handwritten Arabic Documents based on Individual Character Shapes
Manuscript ID: PeerJ 67980 R1
Reviewer Number: 1
Submission Date: Wednesday, March 16, 2022
Thanks to the authors for updating the manuscript. After checking their responses to my comments, I can declare that they have made a suitable major revision.
However, I have two minor comments:
(1) Increase the size (i.e., width) of Figure 7 and Figure 9.
(2) The authors should get another editing help from someone with full professional proficiency in English. For example, Table 2 caption “List of Symbols used” should be “List of Symbols”.

For the authors in case of the authors got a chance to review the manuscript and submit the revised one after the editor’s decision, please, provide a table in the revised manuscript mentioning (1) the comment, (2) the authors’ response, and (3) the authors’ change (if applicable). Please, consider all of the comments and don’t ignore any of them.
Please, refer to the attached file "PeerJ 67980 R1 Reviewer.pdf" for the same comments in an organized format.

---

## Round 0.3 · accepted · Accept

This paper can be accepted. Congratulations.

·

Basic reporting

Thanks to the authors for updating the manuscript. After checking their responses to my comments, I can recommend the acceptance of the manuscript in its current version.

Experimental design

Thanks to the authors for updating the manuscript. After checking their responses to my comments, I can recommend the acceptance of the manuscript in its current version.

Validity of the findings

Thanks to the authors for updating the manuscript. After checking their responses to my comments, I can recommend the acceptance of the manuscript in its current version.

Additional comments

Thanks to the authors for updating the manuscript. After checking their responses to my comments, I can recommend the acceptance of the manuscript in its current version.